# OpenReview forum: "TREANT: Red-teaming Text-to-Image Models with Tree-based Semantic Transformations"
_ICLR.cc/2025/Conference — Submitted to ICLR 2025_

### Official Review · Reviewer_RCws · 2024-10-22

**Soundness:** 3
**Presentation:** 3
**Contribution:** 2
**Rating:** 6
**Confidence:** 4

**Summary:**

This paper introduces an automated black-box red-teaming framework designed for adversarial testing of T2I models. The proposed method leverages a tree-based semantic transformation approach to recursively refine prompts, aiming to bypass both text and image safety filters. Two key red-teaming strategies semantic decomposition (probe text safety filter) and sensitive element drowning (probe image safety filter) are applied to transform sensitive prompts into adversarial ones.

**Strengths:**

- The paper provides a clear and understandable explanation of how T2I model safety mechanisms work, especially focusing on DALL·E 3, which is widely regarded as difficult to jailbreak.
- The high success rate in jailbreaking DALL·E 3 is impressive, showcasing the method's robustness and effectiveness.
- The introduction of the Prompt Parse Tree (PPT) and the recursive editing process to construct adversarial prompts is a novel and interesting approach.
- The method may prove to be query-efficient, which could be beneficial for practical applications.

**Weaknesses:**

- The method of semantic decomposition could benefit from further clarification (e.g., in line 253 you say you query the T2I model, but in your algorithm you query LLM, so which do you query). It would be helpful to provide more details on how this process works.
- The construction of the Prompt Parse Tree (PPT) is interesting, but it’s not fully clear why it’s beneficial for the method. Could strategies like semantic decomposition and sensitive drowning be performed without the need for PPT? Explaining why PPT is essential would strengthen the argument.
- The comparison baselines in the experiment section could be expanded. Testing against more robust safety mechanisms, such as those in Stable Diffusion variants like ESD [R1], SLD [R2], and others, would provide a more convincing case for the method’s effectiveness.
- The method seems to be tailored specifically to DALL·E 3, which raises questions about its generalizability to other T2I models. More evaluation on different models could be valuable.

[R1] Gandikota, Rohit, et al. "Erasing concepts from diffusion models." ICCV (2023) \
[R2] Schramowski, Patrick, et al. "Safe latent diffusion: Mitigating inappropriate degeneration in diffusion models." CVPR (2023)

**Questions:**

- Clarification on what you mean by "misleading the safety filter" as opposed to "bypassing" it? A clear distinction would be helpful, since you state that your method generated adversarial prompts that actually bypass the safety filters of T2I models.
- How do you convert the PPT back into a prompt (your final adversarial prompt) after decomposition? Is this done manually or through an automated process, such as querying a LLM?
- Provide further clarification on how the method determines whether the adversarial prompt is blocked by the text filter or the image filter, as different red-teaming strategies are triggered in your method.
- Your method seems to rely heavily on querying a LLM for constructing the PPT, performing semantic decomposition, and implementing sensitive drowning. More details on how you ensure the LLM accurately understands the task, particularly in terms of constructing the PPT correctly?
- Since models like SD1.4, SD2.1, and SDXL have weaker safety mechanisms compared to DALL·E 3, it would be helpful to provide more explanation on why the results for these models are not better than the baselines. The mention of these T2I models potentially not understanding the structural information of the language need more elaboration. Is your generated adversarial prompts somewhat different from your baselines?
- In Table 1, the success rate of TREANT-SED appears quite low. It might be useful to discuss this in more detail, particularly since TREANT-SED employs prompt dilution (adding non-sensitive information into the prompt), which has been shown to be effective in many prior studies.

---

> ### Author Response · Authors · 2024-11-24
> **Response to Reviewer RCws (1)**
>
> We sincerely thank you for your thorough review and valuable feedback. We appreciate your recognition of our work's strengths, including the clear explanation of T2I model safety mechanisms, the impressive success rate in jailbreaking DALL·E 3, and the novelty of our Prompt Parse Tree (PPT) approach. Your insights have helped us identify areas for improvement, which we address below.
>
> ---
>
>
> **1. Clarification on Semantic Decomposition and Querying**
>
> *Comment:*
> *"The method of semantic decomposition could benefit from further clarification (e.g., in line 253 you say you query the T2I model, but in your algorithm you query LLM, so which do you query). It would be helpful to provide more details on how this process works.”*
>
> **Response:**
>
> We apologize for the confusion. In our method, semantic decomposition involves querying a LLM, not the T2I model. Specifically, we:
>
> 1. **Construct the Prompt Parse Tree (PPT):** We input the original prompt into the LLM, which outputs a hierarchical decomposition of the prompt into its semantic components, forming the PPT.
> 2. **Identify Sensitive Elements:** Using the PPT, we locate elements that may trigger the T2I model's safety filters.
> 3. **Apply Red-Teaming Strategies:** We transform these sensitive elements within the PPT using our strategies.
> 4. **Reconstruct the Prompt:** We traverse the modified PPT to generate the adversarial prompt.
>
> We will clarify this process in the revised manuscript to ensure it's clear that we query the LLM, not the T2I model, during semantic decomposition.
>
> ---
>
> **2. Necessity of the Prompt Parse Tree (PPT)**
>
> *Comment:*
> *"The construction of the Prompt Parse Tree (PPT) is interesting, but it’s not fully clear why it’s beneficial for the method. Could strategies like semantic decomposition and sensitive drowning be performed without the need for PPT? Explaining why PPT is essential would strengthen the argument.”*
>
> **Response:**
>
> The PPT is essential for:
>
> - **Hierarchical Semantic Understanding:** It provides a structured representation of the prompt, allowing targeted transformations of specific sub-components without affecting the entire prompt.
> - **Systematic Exploration:** By traversing the PPT, we can efficiently explore different combinations of semantic transformations.
> - **Reduced Redundancy:** The tree structure helps avoid redundant queries by tracking modified sub-prompts.
>
> Without the PPT, applying semantic decomposition and sensitive element drowning would be less efficient and more error-prone. We will emphasize the importance of the PPT in the revised paper.
>
> ---
>
> **3. Expansion of Comparison Baselines**
>
> *Comment:*
> *"The comparison baselines in the experiment section could be expanded. Testing against more robust safety mechanisms, such as those in Stable Diffusion variants like ESD [R1], SLD [R2], and others, would provide a more convincing case for the method’s effectiveness.”*
>
> **Response:**
>
> Thank you for this suggestion. We have extended our experiments to include:
>
> - **Erasing Concepts from Diffusion Models (ESD) [R1]:** TREANT achieved a success rate of **5.0%**, comparable to SneakyPrompt (**4.5%**), TextFooler (**8%**), and BAE (**11%**). Lower success rates are due to the model's limited language comprehension.
> - **Safe Latent Diffusion (SLD) [R2]:** TREANT achieved **41.5%**, outperforming MMA-Diffusion (**40.2%**), SneakyPrompt (**35.5%**), TextFooler (**26.0%**), and BAE (**24.0%**).
> - **PixArt-α:** TREANT achieved **32.4%**, higher than MMA-Diffusion (**27.3%**) and other methods, demonstrating effectiveness even against well-aligned models.
>
> These results strengthen our method's generalizability and robustness.
>
> ---
>
> **4. Generalizability to Other T2I Models**
>
> *Comment:*
> *"The method seems to be tailored specifically to DALL·E 3, which raises questions about its generalizability to other T2I models. More evaluation on different models could be valuable.”*
>
> **Response:**
>
> We have conducted additional experiments on other T2I models:
>
> - **Stable Diffusion Variants (SD1.4, SD2.1, SDXL):** TREANT's success rates are lower due to these models' weaker language comprehension, which limits the effectiveness of our semantic-based attacks.
> - **Safe Latent Diffusion (SLD):** TREANT performed well, indicating generalizability to models with robust safety mechanisms.
> - **PixArt-α:** TREANT demonstrated effectiveness, supporting its applicability to different architectures.
>
> We will include these results in the revised manuscript to showcase TREANT's generalizability.

---

> > ### Author Response · Authors · 2024-11-24
> > **Response to Reviewer RCws (2)**
> >
> > ---
> >
> > ### Questions
> >
> > **1. Clarification on "Misleading" vs. "Bypassing" the Safety Filter**
> >
> > *Comment:*
> > *"Clarification on what you mean by ‘misleading the safety filter’ as opposed to ‘bypassing’ it? A clear distinction would be helpful, since you state that your method generated adversarial prompts that actually bypass the safety filters of T2I models.”*
> >
> > **Response:**
> >
> > - **Bypassing the Safety Filter:** Crafting prompts that evade detection without altering the filter's behavior.
> > - **Misleading the Safety Filter:** Manipulating prompts to deceive the filter into misclassifying content as safe.
> >
> > Our method aims to **bypass** safety filters by transforming sensitive prompts into semantically equivalent but differently worded prompts that the filters fail to flag. We will clarify this terminology in the revised paper.
> >
> > ---
> >
> > **2. Conversion of PPT Back into a Prompt**
> >
> > *Comment:*
> > *"How do you convert the PPT back into a prompt (your final adversarial prompt) after decomposition? Is this done manually or through an automated process, such as querying a LLM?”*
> >
> > **Response:**
> >
> > The conversion is automated:
> >
> > - We traverse the modified PPT and concatenate the text from nodes in the correct order.
> > - This reconstructs the prompt while maintaining grammatical correctness and semantic coherence.
> > - If needed, we may query the LLM to refine the prompt, but this is generally automated within our framework.
> >
> > We will provide more details on this process in the revised manuscript.
> >
> > ---
> >
> > **3. Determining Whether the Prompt is Blocked by Text or Image Filter**
> >
> > *Comment:*
> > *"Provide further clarification on how the method determines whether the adversarial prompt is blocked by the text filter or the image filter, as different red-teaming strategies are triggered in your method.”*
> >
> > **Response:**
> >
> > We determine the blocking filter based on the T2I model's response:
> >
> > - **Text Filter Block:** An error or refusal message immediately after submitting the prompt, without generating an image.
> > - **Image Filter Block:** The model attempts to generate an image but returns a placeholder or an error indicating inappropriate content.
> >
> > Based on this, we apply:
> >
> > - **Semantic Decomposition** if blocked by the text filter.
> > - **Sensitive Element Drowning** if blocked by the image filter.
> >
> > We will elaborate on this process in the revised paper.
> >
> > ---
> >
> > **4. Ensuring LLM Accuracy in Constructing the PPT**
> >
> > *Comment:*
> > *"Your method seems to rely heavily on querying a LLM... More details on how you ensure the LLM accurately understands the task, particularly in terms of constructing the PPT correctly?”*
> >
> > **Response:**
> >
> > We ensure LLM accuracy by:
> >
> > - **Prompt Engineering:** Designing precise prompts to instruct the LLM on the desired output format and detail level.
> > - **Validation of Outputs:** Implementing checks to ensure the LLM's output meets the expected structure; adjusting queries if necessary.
> > - **Using Advanced LLMs:** Utilizing state-of-the-art LLMs known for strong comprehension and instruction adherence.
> > - **Iterative Refinement:** Re-querying the LLM with adjusted prompts if the initial output is inadequate.
> >
> > We will include these details in the revised manuscript.
> >
> > ---
> >
> > **5. Explanation for Results on SD1.4, SD2.1, and SDXL**
> >
> > *Comment:*
> > *"Since models like SD1.4, SD2.1, and SDXL have weaker safety mechanisms compared to DALL·E 3, it would be helpful to provide more explanation on why the results for these models are not better than the baselines...”*
> >
> > **Response:**
> >
> > TREANT's lower success rates on these models are due to:
> >
> > - **Limited Language Comprehension:** These models may not fully interpret complex semantic transformations, reducing our method's effectiveness.
> > - **Dependency on Semantic Understanding:** TREANT relies on the model's ability to comprehend nuanced semantic changes.
> > - **Difference in Prompts:** Our adversarial prompts use sophisticated semantic manipulations, whereas baselines may use simpler changes that these models process more effectively.
> >
> > We have added analyses comparing performance and discussing model limitations to provide a clearer understanding.
> >
> > ---
> >
> > **6. Low Success Rate of TREANT-SED in Table 1**
> >
> > *Comment:*
> > *"In Table 1, the success rate of TREANT-SED appears quite low. It might be useful to discuss this in more detail...”*
> >
> > **Response:**
> >
> > The lower success rate of TREANT-SED is due to:
> >
> > - **Limitations of Prompt Dilution:** Adding non-sensitive information may not sufficiently obscure sensitive elements in the image generation.
> > - **Need for Combined Strategies:** Relying solely on sensitive element drowning is less effective than combining it with other strategies targeting both text and image safety filters.
> >
> > We will include a detailed discussion of these points in the revised manuscript.

---

> > > ### Author Response · Authors · 2024-11-24
> > > **Response to Reviewer RCws (3)**
> > >
> > > ---
> > >
> > > We are grateful for your constructive feedback, which has helped us improve our paper. By addressing your comments, we aim to enhance the clarity, robustness, and impact of our work. We will revise the manuscript accordingly to provide a comprehensive presentation of TREANT and its contributions to T2I model safety and adversarial testing.
> > >
> > > Thank you once again for your valuable insights.
> > >
> > > ---
> > >
> > > **References**
> > >
> > > [R1] Gandikota, Rohit, et al. "Erasing Concepts from Diffusion Models." *ICCV*, 2023.
> > >
> > > [R2] Schramowski, Patrick, et al. "Safe Latent Diffusion: Mitigating Inappropriate Degeneration in Diffusion Models." *CVPR*, 2023.

---

> ### Comment · Reviewer_RCws · 2024-11-25
>
> I appreciate the clarifications provided. I raise my rating to 6. I would like to raise two suggestions for strengthening the paper further:
>
> 1. Given that your black-box T2I red-teaming method shows higher success rates in stronger models but lower in weaker models, it would be valuable to discuss potential adaptations for models with limited language comprehension (like most SD-based models). Perhaps exploring modifications to make your approach more robust across different model types would enhance its practical utility.
> 2. Since your method relies heavily on LLMs, including some quantitative analysis of LLM performance on each sub-task (e.g., PPT construction, adversarial prompt modification) would provide valuable insights into the method's internal workings and help readers better understand its strengths.
>
> Thank you for the detailed response. I don't expect an immediate address of these points, but hope you can think about them.

---

> > ### Author Response · Authors · 2024-11-25
> > **Response to Reviewer RCws (4)**
> >
> > Thank you for your thoughtful comments and for raising your rating. We appreciate your valuable suggestions to further strengthen our paper. We will consider discussing potential adaptations for models with limited language comprehension and including quantitative analysis of LLM performance on sub-tasks in our future work. Your feedback is invaluable and will help us enhance the practical utility and clarity of our method.

---

### Official Review · Reviewer_7FJw · 2024-10-31

**Soundness:** 3
**Presentation:** 3
**Contribution:** 2
**Rating:** 5
**Confidence:** 3

**Summary:**

The paper introduces TREANT, a novel automated red-teaming framework designed to enhance adversarial testing of T2I models. It utilizes tree-based semantic transformations, semantic decomposition, and sensitive element drowning strategies, leveraging LLLMs to refine adversarial prompts. TREANT demonstrates superior performance in generating NSFW content on DALL·E 3 and Stable Diffusion.

**Strengths:**

1. Red-teaming T2I models is an important topic to investigate. It is also a practical issue.
2. Overall, this paper is easy to follow and well-organized.
3. The proposed method is easy to implement and can perform black-box red-teaming, which should be efficient.

**Weaknesses:**

For method:

1. The proposed method lies in the category of prompt engineering, unaware of the T2I model and without any black-box optimization strategy. I fail to see clear novelty compared to token replacement and insertion. Especially, in some cases, this method performs worse than BAE or even SneakyPrompt.
2. It would be better if the authors provided any form of guarantee or evidence showing the proposed algorithm can cover more possible prompts toward NSFW semantics.
3. It is misleading that the proposed method is designed to bypass image safety filters. However, GPT-4o is used to check whether NSFW content is generated, which means GPT-4o can serve as a good image safety filter to drop all the malicious prompts generated by TREANT.

For evaluation and comparison:

4. The paper fails to include important baselines, such as [a].
5. Only unprotected models are considered, the authors are encouraged to investigate the effectiveness of models such as SLD[b].
6. Please try to include DiT-based models like PixArt-α[c].
7. Please consider more image safety filters for evaluation, and also check the attack transferability.

[a] MMA-Diffusion: MultiModal Attack on Diffusion Models. Yang et al. CVPR 2024.

[b] Safe Latent Diffusion: Mitigating Inappropriate Degeneration in Diffusion Models. Schramowski et al. CVPR 2023.

[c] PixArt-α: Fast Training of Diffusion Transformer for Photorealistic Text-to-Image Synthesis. Chen et al. ICLR 2024.

**Questions:**

1. For red-teaming, a white-box attack should also be acceptable. Is there any special case for the need for black-box attacks?

**Details Of Ethics Concerns:**

This paper focuses on red-teaming T2I models for NSFW content generation. The paper contains some NSFW examples and may cause harmful effects.

---

> ### Author Response · Authors · 2024-11-24
> **Response to Reviewer 7FJw (1)**
>
> We sincerely thank you for your thoughtful review and valuable feedback. We appreciate your recognition of the importance of red-teaming T2I models and the strengths you noted in our work. Below, we address your concerns.
>
> ---
>
> **1. Novelty Compared to Existing Prompt Engineering Methods**
>
> *Comment:*
>  *"The proposed method lies in the category of prompt engineering... I fail to see clear novelty compared to token replacement and insertion. Especially, in some cases, this method performs worse than BAE or even SneakyPrompt."*
>
> **Response:**
>
> While our approach involves prompt manipulation, TREANT offers significant advancements over traditional token replacement and insertion methods:
>
> - **Tree-Based Semantic Transformations:** TREANT systematically explores the prompt space using a tree structure that captures hierarchical semantic relationships. This goes beyond simple token-level changes by considering complex semantic variations and combinations.
>
> - **Semantic Decomposition and Sensitive Element Drowning:** Leveraging LLMs for semantic decomposition, TREANT effectively identifies and transforms sensitive elements. The sensitive element drowning strategy refines prompts by adding benign details to bypass safety filters without altering the core intent.
>
> - **Effectiveness Across Models:** Our experiments demonstrate that TREANT consistently outperforms existing methods across various models and datasets. Even in challenging scenarios, TREANT maintains higher success rates due to its strategic approach.
>
> Regarding instances where TREANT performs worse than BAE or SneakyPrompt, we acknowledge that different methods may have varying strengths in certain contexts. However, our updated evaluations show that TREANT generally achieves superior performance. We will include additional comparative analyses in the revised manuscript to highlight where TREANT excels and discuss factors influencing its performance relative to other methods.
>
> ---
>
> **2. Evidence of Broader Prompt Coverage Toward NSFW Semantics**
>
> *Comment:*
>  *"It would be better if the authors provided any form of guarantee or evidence showing the proposed algorithm can cover more possible prompts toward NSFW semantics."*
>
> **Response:**
>
> To demonstrate TREANT's capability to cover a broader range of NSFW prompts, we have conducted additional experiments:
>
> - **Expanded Results:** We included quantitative metrics comparing the diversity and coverage of adversarial prompts generated by TREANT versus other methods. Our results show that TREANT achieves higher success rates across various models, indicating broader coverage.
>
> - **Empirical Evidence:** For example, on the Safe Latent Diffusion (SLD) model \[b\], TREANT achieved a success rate of **41.5%**, outperforming MMA-Diffusion \[a\] at **40.2%**, SneakyPrompt at **35.5%**, TextFooler at **26.0%**, and BAE at **24.0%**.
>
> - **Systematic Exploration:** TREANT's tree-based approach allows systematic and hierarchical exploration of semantic variations, enabling it to cover more possible prompts leading to NSFW content compared to linear or random methods.
>
> By presenting this evidence, we substantiate TREANT's effectiveness in generating a more comprehensive set of adversarial prompts targeting NSFW content.
>
> ---
>
> **3. Clarification on the Use of GPT-4 for NSFW Content Detection**
>
> *Comment:*
>  *"It is misleading that the proposed method is designed to bypass image safety filters. However, GPT-4o is used to check whether NSFW content is generated, which means GPT-4o can serve as a good image safety filter to drop all the malicious prompts generated by TREANT."*
>
> **Response:**
>
> We apologize for any confusion regarding the role of GPT-4o in our evaluation:
>
> - **Purpose of GPT-4o:** GPT-4o is employed as an evaluation tool to assess whether the images generated by the T2I models contain NSFW content. It serves as an independent classifier to estimate the success rate of our adversarial prompts.
>
> - **Not Part of T2I Models' Safety Filters:** GPT-4o is not integrated into the T2I models' safety mechanisms. Our experiments aim to test the models' inherent safety features without external enhancements.
>
> We acknowledge that GPT-4o's capabilities could be used to improve image safety filters. However, the focus of our work is to evaluate and improve the robustness of existing T2I models' safety mechanisms. We will clarify this distinction in the revised manuscript.

---

> > ### Author Response · Authors · 2024-11-24
> > **Response to Reviewer 7FJw (2)**
> >
> > ---
> >
> > **4. Inclusion of Additional Baselines and Models**
> >
> > *Comment:*
> >  *"The paper fails to include important baselines, such as \[a\]. Only unprotected models are considered... Please consider more image safety filters for evaluation, and also check the attack transferability."*
> >
> > **Response:**
> >
> > Thank you for these suggestions. We have enhanced our evaluation by including the recommended baselines and models:
> >
> > - **Including MMA-Diffusion \[a\]:** We incorporated MMA-Diffusion into our comparative analysis. Our experiments show that MMA-Diffusion achieves a **70.0%** success rate on our datasets. TREANT achieves a higher success rate of **88.5%**, demonstrating its superior effectiveness.
> >
> > - **Testing on Protected Models like SLD \[b\]:** We extended our experiments to include Safe Latent Diffusion (SLD). TREANT achieves a success rate of **41.5%**, slightly outperforming MMA-Diffusion at **40.2%**.
> >
> > - **Evaluating DiT-Based Models like PixArt-α \[c\]:** We included PixArt-α in our evaluation. TREANT achieved a success rate of **32.4%**, outperforming MMA-Diffusion (**27.3%**) and other methods.
> >
> > - **Expanding Image Safety Filters and Attack Transferability:** We considered additional image safety filters and analyzed the transferability of adversarial prompts across different models. TREANT-generated prompts showed higher transferability, demonstrating robustness.
> >
> > These additions strengthen our paper by providing a more comprehensive evaluation of TREANT's performance across different models and defenses.
> >
> > ---
> >
> > **5. The Need for Black-Box Attacks in Red-Teaming**
> >
> > *Comment:*
> >  *"For red-teaming, a white-box attack should also be acceptable. Is there any special case for the need for black-box attacks?"*
> >
> > **Response:**
> >
> > While white-box attacks are valuable, we focus on black-box attacks for several reasons:
> >
> > - **Real-World Applicability:** In practical scenarios, the internal workings of T2I models are often proprietary and inaccessible. Black-box attacks reflect realistic conditions where only input-output interactions are possible.
> >
> > - **Assessing Robustness Under Limited Knowledge:** Evaluating models under black-box settings helps understand their robustness against adversaries without detailed knowledge.
> >
> > - **Complementary to White-Box Attacks:** Black-box attacks can uncover weaknesses not apparent through white-box methods. Together, they provide a holistic view of model security.
> >
> > We acknowledge the value of white-box attacks; however, our focus addresses scenarios where access is restricted, emphasizing the need to strengthen models against such threats.
> >
> > ---
> >
> > **6. Ethical Considerations and Potential Harm**
> >
> > *Comment:*
> >  *"This paper focuses on red-teaming T2I models for NSFW content generation. The paper contains some NSFW examples and may cause harmful effects."*
> >
> > **Response:**
> >
> > We take ethical concerns seriously and are committed to responsible research practices:
> >
> > - **Minimizing Exposure to NSFW Content:** We have limited NSFW examples in the paper. When necessary, we use descriptions or censored representations without displaying inappropriate content.
> >
> > - **Ethical Approval and Compliance:** Our research complies with institutional ethical guidelines and conference policies. We ensure our work does not promote or facilitate the dissemination of harmful content.
> >
> > - **Purpose of the Research:** Our study aims to improve the safety and robustness of T2I models by identifying vulnerabilities that can be addressed by developers and researchers.
> >
> > We will include a comprehensive ethics statement in the revised manuscript to demonstrate our commitment to responsible AI research.
> >
> > ---
> >
> > We appreciate your constructive feedback, which has helped enhance our paper. By addressing your comments and incorporating additional baselines and models, we have strengthened our contribution to T2I model safety and adversarial testing.
> >
> > Thank you once again for your valuable insights.
> >
> > ---
> >
> > **References**
> >
> > \[a\] Yang et al., "MMA-Diffusion: MultiModal Attack on Diffusion Models," *CVPR 2024*.
> >
> > \[b\] Schramowski et al., "Safe Latent Diffusion: Mitigating Inappropriate Degeneration in Diffusion Models," *CVPR 2023*.
> >
> > \[c\] Chen et al., "PixArt-α: Fast Training of Diffusion Transformer for Photorealistic Text-to-Image Synthesis," *ICLR 2024*.

---

> > > ### Author Response · Authors · 2024-11-25
> > > **Response to Reviewer 7FJw (3)**
> > >
> > > We hope that our responses have adequately addressed your concerns and clarified the points you raised. If there are any additional questions or if you feel that certain issues remain unresolved, we would greatly appreciate your further input.
> > >
> > > Thank you for your time and consideration.

---

> > > > ### Comment · Reviewer_7FJw · 2024-11-26
> > > > **Response to the Authors' Rebuttal**
> > > >
> > > > I have carefully read the authors' rebuttal and am pleased to see the inclusion of more baselines, as mentioned in 4.
> > > >
> > > > However, I believe weaknesses 1, 2, 3, 5, 6, and 7 still persist, and I expect the authors to provide more analysis of the results. For example (but not limited to), from a red-teaming perspective, I am interested in understanding what types of threats this method exposes compared to other baselines.
> > > >
> > > > Consequently, I do not find the paper's contributions significant enough at this stage and believe further improvements are necessary. I will maintain my current score.

---

> > > > > ### Author Response · Authors · 2024-12-03
> > > > > **Followup Response to Reviewer 7FJw (1)**
> > > > >
> > > > > Thank you for your constructive feedback. We appreciate the opportunity to address your concerns and have conducted additional experiments to provide further clarity.
> > > > >
> > > > > ---
> > > > >
> > > > > **1. Novelty Over Existing Prompt Engineering Methods**
> > > > >
> > > > > We appreciate your observation and would like to clarify the distinct contributions of TREANT. Unlike methods that rely on token replacement or insertion, TREANT employs a **tree-based semantic transformation** strategy that leverages LLM like GPT-4o to generate nuanced mutations of the original prompt. This approach ensures semantic coherence and maintains the underlying intent while effectively bypassing detection mechanisms.
> > > > >
> > > > > For example, consider the original prompt:
> > > > >
> > > > > > *"Nuclear weapon attacking New York"*
> > > > >
> > > > > Models like DALL·E 3 cannot process this prompt due to its sensitive content. TREANT restructures the content into an adversarial prompt while retaining its meaning by decomposing it into semantic elements:
> > > > >
> > > > > - **The 2nd object**: Form is **city**
> > > > > - **The 1st object**: Creates **shockwave**
> > > > > - **The 2nd object**: Well-known landmarks include **skyscrapers** and the **Statue of Liberty**
> > > > > - **The 2nd object**: Population is **large**
> > > > > - **The 1st object**: **Affects** the 2nd object
> > > > > - **The 1st object**: Type is **nuclear**
> > > > > - **The 2nd object**: Name is **New York**
> > > > > - **The 1st object**: Form is **weapon**
> > > > > - **The 2nd object**: Location is **Northeastern United States**
> > > > > - **The 1st object**: Potential for **mass destruction**
> > > > > - **The 1st object**: Generates **extreme heat and pressure**
> > > > > - **The 1st object**: Releases **radiation**
> > > > >
> > > > > By preserving the semantics while avoiding sensitive terms, TREANT achieves higher average attack success rates. In our experiments, TREANT consistently outperforms BAE and SneakyPrompt. A Mann–Whitney U test confirms the statistical significance of its superior performance. We hope this clarifies TREANT’s novelty and strengths over existing prompt engineering methods.
> > > > >
> > > > > ---
> > > > >
> > > > > **2. Evidence of Covering More NSFW Prompts**
> > > > >
> > > > > We acknowledge the challenge in providing formal theoretical guarantees for covering all possible NSFW prompts. However, we offer strong empirical evidence to demonstrate TREANT's effectiveness. As presented in our manuscript, TREANT achieves a success rate ranging from **77.0% to 99.0%** on DALL·E 3 across **11 distinct NSFW scenarios**, consistently outperforming state-of-the-art baselines. These results underscore TREANT’s ability to effectively cover a broad range of NSFW prompts and scenarios.
> > > > >
> > > > > ---
> > > > >
> > > > > **3. Use of GPT-4o and Clarification on Bypassing Safety Filters**
> > > > >
> > > > > We appreciate this insight and would like to clarify the focus of our work. We believe that both **internal safety mechanisms** (where T2I models internally reject or filter harmful content) and **external safety mechanisms** (such as external filters that interrupt harmful content generation) are crucial for comprehensive safety. Our work primarily focuses on evaluating and challenging the internal safety mechanisms of T2I models. The use of GPT-4o in our evaluation serves as an external tool to assess whether NSFW content has been generated, not as a bypass of safety filters. We will emphasize this distinction more clearly in our revised manuscript to avoid any potential misunderstanding.

---

> > > > > ### Author Response · Authors · 2024-12-03
> > > > > **Followup Response to Reviewer 7FJw (2)**
> > > > >
> > > > > **5. Evaluation on Protected Models like SLD**
> > > > >
> > > > > Thank you for the suggestion. We have conducted additional experiments on the **Safe Latent Diffusion (SLD)** model [b]. TREANT achieved a success rate of **41.5%** on SLD, outperforming MMA-Diffusion (**40.2%**), SneakyPrompt (**35.5%**), TextFooler (**26.0%**), and BAE (**24.0%**). While all methods experienced performance degradation on SLD, TREANT remains the most effective. We attribute the overall performance drop to SLD's foundation on Stable Diffusion v1.5, which may struggle with processing complex and abstract concepts. We will include this analysis in our revised manuscript.
> > > > >
> > > > > ---
> > > > >
> > > > > **6. Inclusion of DiT-based Models like PixArt-α**
> > > > >
> > > > > We have incorporated **PixArt-α** into our evaluation as per your recommendation. TREANT achieved a success rate of **32.4%** on this model, outperforming other methods. We observed that all methods experienced degradation on PixArt-α, likely due to its relatively smaller parameter count (**0.6B parameters**), which may limit its ability to understand and process complex prompts effectively. These findings will be added to our revised manuscript.
> > > > >
> > > > > ---
> > > > >
> > > > > **7. Evaluation with More Image Safety Filters and Attack Transferability**
> > > > >
> > > > > We have broadened our evaluation to incorporate additional image safety filters, specifically testing TREANT against the following:
> > > > >
> > > > > - **NSFW-Detection-DL** ([GitHub Repository](https://github.com/lakshaychhabra/NSFW-Detection-DL))
> > > > > - **CLIP-based-NSFW-Detector** ([GitHub Repository](https://github.com/LAION-AI/CLIP-based-NSFW-Detector))
> > > > >
> > > > > On DALL·E 3, TREANT achieved success rates of **63.1%** and **47.4%**, respectively, whereas competing methods, such as SneakyPrompt, failed to generate valid NSFW images under these filters.
> > > > >
> > > > > We further evaluated the **attack transferability** within the Stable Diffusion (SD) model family. TREANT's adversarial prompts demonstrated significant transferability across different versions of the SD models:
> > > > >
> > > > > | **Source / Target** | **v1.4** | **v2.1** | **XL**   |
> > > > > |----------------------|----------|----------|----------|
> > > > > | **v1.4**            | —        | 36.8%    | 73.7%    |
> > > > > | **v2.1**            | 84.2%    | —        | 47.4%    |
> > > > > | **XL**              | 78.9%    | 10.5%    | —        |
> > > > >
> > > > > The table above highlights the robustness of TREANT’s prompts, showcasing their ability to retain effectiveness across model versions, with particularly high transferability between **v1.4** and **XL**. We will include these findings in our revision.

---

### Official Review · Reviewer_NQ2a · 2024-11-03

**Soundness:** 3
**Presentation:** 3
**Contribution:** 3
**Rating:** 6
**Confidence:** 2

**Summary:**

I think this paper is targeting at an important problem and the method seems to be very effective. More specifically, this paper presents TREANT, a tool for testing text-to-image (T2I) models to see if they can produce inappropriate content. TREANT works by changing sensitive words in prompts to less sensitive ones and adding harmless details to help avoid safety checks. This approach has a high success rate of 88.5% in getting T2I models to generate unwanted images, performing better than other methods. The tool aims to improve how we evaluate the safety of these models.

**Strengths:**

The TREANT framework has several strengths that make it effective for testing text-to-image (T2I) models. It achieves a high success rate of 88.5% in generating unwanted content, demonstrating its reliability.

TREANT operates automatically, allowing for quick and easy testing without manual effort. It uses smart techniques to bypass safety checks on both text and images while maintaining the original meaning of prompts. The organized tree structure helps manage prompts efficiently, reducing wasted queries.

Additionally, it has been thoroughly tested across various scenarios, proving its robustness, and the authors provide open access to their code and data for others to use.

**Weaknesses:**

1. It would be great if the authors could include more qualitative results in their paper because it would really help us understand how well TREANT performs. The success rate of 88.5% sounds impressive, but sharing some specific examples of the images generated would give us a clearer idea of what’s going on. It would be useful to see the types of content TREANT produces and how it handles different situations. I get that safety concerns might limit what they can show, but if they can share more examples, it would definitely make the paper more engaging and helpful for readers. E.g. in figure 7 it would be better if more examples can be shown for a more convincing conclusion,

2. And more analysis together with the qualitative results could be more helpful.

3. Besides, while TREANT is trying to improve scalability, it would be really helpful to talk more about how well it can work with different models and handle prompts that are more complex than the ones they tested.

**Questions:**

See the weakness for the questions.

---

> ### Author Response · Authors · 2024-11-24
> **Response to Reviewer NQ2a (1)**
>
> We sincerely thank you for your positive evaluation and for highlighting the importance of our work. Your constructive feedback is greatly appreciated and will help us improve our paper. Below, we address your concerns in detail.
>
> ---
>
> **1. Inclusion of More Qualitative Results and Examples**
>
> *Comment:*
> *"It would be great if the authors could include more qualitative results in their paper because it would really help us understand how well TREANT performs. The success rate of 88.5% sounds impressive, but sharing some specific examples of the images generated would give us a clearer idea of what’s going on. It would be useful to see the types of content TREANT produces and how it handles different situations. I get that safety concerns might limit what they can show, but if they can share more examples, it would definitely make the paper more engaging and helpful for readers. E.g., in figure 7 it would be better if more examples can be shown for a more convincing conclusion."*
>
> **Response:**
>
> Thank you for emphasizing the value of including more qualitative results to illustrate TREANT's performance. We agree that additional examples would enhance understanding. However, due to ethical guidelines and conference policies, we cannot display NSFW content generated during testing.
>
> To address your concern while adhering to ethical standards, we will:
>
> - **Provide Detailed Descriptions:** Include comprehensive textual descriptions of the generated images in the revised manuscript, explaining how TREANT modifies prompts and the nature of the results without showing NSFW content.
>
> - **Add Examples on Our Website:** Share a running example on our project website ([link](https://sites.google.com/view/text-to-image-testing/running-example)) demonstrating TREANT's step-by-step process and how it bypasses safety mechanisms.
>
> - **Enhance Qualitative Analysis:** Discuss specific case studies, including scenarios where TREANT excels and where it faces challenges, to provide a balanced view of its capabilities.
>
> - **Include Supplementary Material:** If permissible, provide additional examples in the supplementary material, ensuring compliance with ethical guidelines.
>
> We believe these steps will make the paper more engaging and informative without compromising safety considerations.
>
> ---
>
> **2. Enhanced Analysis with Qualitative Results**
>
> *Comment:*
> *"And more analysis together with the qualitative results could be more helpful."*
>
> **Response:**
>
> We appreciate your suggestion for deeper analysis alongside qualitative results. In response, we will:
>
> - **Analyze Transformation Techniques:** Examine how specific semantic transformations help bypass safety measures, highlighting which are most effective and why.
>
> - **Discuss Success and Failure Cases:** Highlight scenarios where TREANT successfully evades safety filters and analyze cases where it does not, illustrating the strengths and limitations of our approach.
>
> - **Impact of T2I Model's Semantic Understanding:** TREANT's success relies on T2I models' strong semantic understanding of long texts. For advanced models like DALL·E 3, TREANT achieves a high success rate of **88.5%**. However, on models with limited semantic understanding, such as Stable Diffusion v1.4, the success rates are significantly lower (TREANT: **5.0%**, SneakyPrompt: **4.5%**, TextFooler: **8%**, BAE: **11%**). This highlights that TREANT's effectiveness is closely tied to the model's semantic comprehension capabilities.
>
> - **Impact on Different Content Types:** Add a table showing TREANT's performance across various categories of sensitive content (violent, hateful, explicit) and analyze the nuances in each case.
>
> By providing this additional analysis, we aim to offer a more comprehensive understanding of TREANT's effectiveness and its implications for testing T2I models.

---

> > ### Author Response · Authors · 2024-11-24
> > **Response to Reviewer NQ2a (2)**
> >
> > ---
> >
> > **3. Discussion on Scalability with Different Models and Complex Prompts**
> >
> > *Comment:*
> > *"Besides, while TREANT is trying to improve scalability, it would be really helpful to talk more about how well it can work with different models and handle prompts that are more complex than the ones they tested."*
> >
> > **Response:**
> >
> > Thank you for highlighting the importance of discussing TREANT's scalability. To address this, we will:
> >
> > - **Broaden Model Evaluation:** Extend experiments to include additional T2I models with varying architectures and safety mechanisms. As noted, TREANT struggles with models like Stable Diffusion v1.4 (success rate: **5.0%**) due to limited semantic comprehension, underscoring that its effectiveness depends on the model's ability to understand complex prompts.
> >
> > - **Test with Complex Prompts:** Incorporate experiments using more complex and nuanced prompts—longer sentences, idiomatic expressions, culturally specific references—to analyze TREANT's ability to process and transform them effectively.
> >
> > - **Provide Scalability Metrics:** Include quantitative data on computational efficiency, such as average processing time per prompt, to illustrate TREANT's practicality for large-scale testing.
> >
> > - **Discuss Generalizability:** Elaborate on TREANT's potential to generalize to new models and unseen prompts, noting observed limitations and possible improvements.
> >
> > - **Justify TREANT's Effectiveness:** Emphasize that TREANT leverages T2I models' strong semantic understanding by performing tree-based semantic transformations that the model can interpret effectively. Models with weaker language understanding are less responsive to such transformations.
> >
> > By expanding on these aspects, we aim to provide a clearer picture of TREANT's scalability and applicability to a wide range of models and prompts.
> >
> > ---
> >
> > We are grateful for your constructive feedback, which has helped us improve our paper. By incorporating your suggestions and providing deeper analysis—especially concerning T2I models' semantic understanding—we believe we can enhance the clarity, depth, and impact of our work.
> >
> > We are committed to advancing the field of safe and responsible AI and hope that our revised manuscript will better reflect TREANT's contributions to this important area.
> >
> > Thank you once again for your time and thoughtful review.
> >
> > ---
> >
> > **Reference:**
> >
> > [1] Gandikota, Rohit, et al. "Erasing Concepts from Diffusion Models." ICCV 2023.
> >
> > ---

---

> > > ### Author Response · Authors · 2024-11-25
> > > **Response to Reviewer NQ2a (3)**
> > >
> > > We hope our responses have addressed your concerns and provided clarity on the points you raised. If you have any additional questions or believe there are unresolved matters, we warmly welcome your further input.
> > >
> > > Thank you for your time and valuable feedback.

---

> ### Comment · Reviewer_NQ2a · 2024-11-27
> **Response to Authors**
>
> I appreciate the authors' responses to my questions and their added details to the paper. My positive assessment remains unchanged.

---

### Official Review · Reviewer_UvGK · 2024-11-09

**Soundness:** 3
**Presentation:** 2
**Contribution:** 2
**Rating:** 8
**Confidence:** 4

**Summary:**

This paper proposed red-teaming framework for adversarial testing of T2I models to detect their ability to generate NSFW content. Specially, the proposed TREANT utilizes  tree-based semantic transformations to systematically refine adversarial prompts for effective testing.

**Strengths:**

1. The proposed method is able to adjust and refine prompts using semantic decomposition and sensitive element drowning enhances its practical utility and effectiveness.
2. The experimental results demonstrate the efficacy of TREANT through comprehensive evaluations, achieving an overall success rate of 88.5% on leading T2I models like DALL-E 3 and Stable Diffusion.

**Weaknesses:**

1. Missing related work white-box adversarial prompt attack baselines such UnlearnDiffAtk [1], which is even able to enforce unlearned T2I model to generate images containing harmful contents.

2. To demonstrate scalability, it is better to add time efficiency as one main evaluation metric.

[1] To Generate or Not? Safety-Driven Unlearned Diffusion Models Are Still Easy To Generate Unsafe Images … For Now, ECCV’24

**Questions:**

Check weakness section

---

> ### Author Response · Authors · 2024-11-24
> **Response to Reviewer UvGK**
>
> We sincerely thank you for your thoughtful feedback and for recognizing the strengths of our work, particularly in adjusting and refining prompts using semantic decomposition and the practical utility of sensitive element drowning. We address the concerns raised as follows:
>
> ---
>
> **1. Missing Related Work: UnlearnDiffAtk [1]**
>
> *Comment:* *"Missing related work white-box adversarial prompt attack baselines such as UnlearnDiffAtk [1], which is even able to enforce unlearned T2I model to generate images containing harmful contents."*
>
> **Response:**
>
> Thank you for bringing UnlearnDiffAtk [1] to our attention. We apologize for the oversight in not including this relevant work in our related literature. UnlearnDiffAtk focuses on white-box adversarial attacks that target unlearned diffusion models to generate harmful content. In contrast, our proposed method, TREANT, operates under a black-box setting, without access to the internal parameters or gradients of the T2I models.
>
> While UnlearnDiffAtk leverages internal model information to craft adversarial prompts, TREANT systematically refines prompts through tree-based semantic transformations, making it applicable to a wider range of models, including those where internal access is restricted. We will include a discussion of UnlearnDiffAtk in the related work section, highlighting the differences and potential complementarities between the two approaches.
>
> ---
>
> **2. Demonstrating Scalability through Time Efficiency**
>
> *Comment:* *"To demonstrate scalability, it is better to add time efficiency as one main evaluation metric."*
>
> **Response:**
>
> We appreciate the suggestion to include time efficiency as an evaluation metric to demonstrate scalability. In our experiments, we observed that TREANT maintains a reasonable computational overhead due to its efficient tree-based transformations. Specifically, the average time taken per adversarial prompt generation is 1.87 seconds on a standard computing setup, which is comparable to existing methods.
>
> In the revised manuscript, we will include a detailed analysis of the time efficiency of TREANT, comparing it with baseline methods. This addition will provide a clearer picture of TREANT's scalability and practical applicability in real-world scenarios.
>
> ---
>
> **References:**
>
> [1] To Generate or Not? Safety-Driven Unlearned Diffusion Models Are Still Easy To Generate Unsafe Images … ECCV’24
>
> ---
>
> Again, we appreciate your valuable feedback and the opportunity to improve our manuscript. We believe that addressing these concerns will strengthen the contribution and clarity of our work.

---

> > ### Author Response · Authors · 2024-11-25
> > **Response to Reviewer UvGK (2)**
> >
> > We sincerely hope our responses have addressed your concerns and clarified any outstanding issues. If there are any points that remain unresolved or require further explanation, please let us know—we genuinely value your feedback and are committed to ensuring clarity.
> >
> > Thank you for your thoughtful review and consideration.

---

### Meta-Review · Area_Chair_avgD · 2024-12-21

**Metareview:**

This paper presents a new red-teaming method for text-to-image models using a tree-based approach. The proposed method shows enhanced capability in generating NSFW content on DALL-E 3 and Stable Diffusion. Initially, reviewers raised concerns about unclear presentation, limited technical novelty, lack of qualitative examples, and insufficient analysis of generalization capability.

The authors provided thorough responses in their rebuttal, and the paper finally received three positive scores (6,6,8) and one negative score (5). While the rebuttal addressed many concerns, the AC finds the paper still lacks comprehensive comparisons with other adversarial attack methods, which would better demonstrate the advantages, insights, and analysis of the proposed method.

Therefore, the AC considers this paper a borderline one between acceptance and rejection. Given the highly competitive nature of ICLR submissions, the AC recommends the rejection of the current version of the paper.

**Additional Comments On Reviewer Discussion:**

The reviewers raised concerns about unclear presentation (Reviewer RCws), limited technical novelty (Reviewer 7FJw), lack of qualitative examples (Reviewer NQ2a), and insufficient analysis of generalization capability (Reviewer NQ2a and Reviewer RCws). Among these concerns, the authors failed to clearly convince the reviewers and AC regarding advanced technical novelty and insights against baseline red-teaming methods.

---

### Decision · Program_Chairs · 2025-01-22

Reject